# Serum Uric Acid Predicts All-Cause and Cardiovascular Mortality Independently of Hypertriglyceridemia in Cardiometabolic Patients without Established CV Disease: A Sub-Analysis of the URic acid Right for heArt Health (URRAH) Study

**DOI:** 10.3390/metabo13020244

**Published:** 2023-02-07

**Authors:** Alessandro Mengozzi, Nicola Riccardo Pugliese, Giovambattista Desideri, Stefano Masi, Fabio Angeli, Carlo Maria Barbagallo, Michele Bombelli, Federica Cappelli, Edoardo Casiglia, Rosario Cianci, Michele Ciccarelli, Arrigo F. G. Cicero, Massimo Cirillo, Pietro Cirillo, Raffaella Dell’Oro, Lanfranco D’Elia, Claudio Ferri, Ferruccio Galletti, Loreto Gesualdo, Cristina Giannattasio, Guido Grassi, Guido Iaccarino, Luciano Lippa, Francesca Mallamaci, Alessandro Maloberti, Maria Masulli, Alberto Mazza, Maria Lorenza Muiesan, Pietro Nazzaro, Paolo Palatini, Gianfranco Parati, Roberto Pontremoli, Fosca Quarti-Trevano, Marcello Rattazzi, Gianpaolo Reboldi, Giulia Rivasi, Elisa Russo, Massimo Salvetti, Valerie Tikhonoff, Giuliano Tocci, Andrea Ungar, Paolo Verdecchia, Francesca Viazzi, Massimo Volpe, Claudio Borghi, Agostino Virdis

**Affiliations:** 1Department of Clinical and Experimental Medicine, University of Pisa, 56126 Pisa, Italy; 2Center for Translational and Experimental Cardiology (CTEC), Department of Cardiology, University Hospital Zurich, University of Zurich, 8952 Schlieren, Switzerland; 3Scuola Superiore Sant’Anna, 56127 Pisa, Italy; 4Department of Life, Health and Environmental Sciences, University of L’Aquila, 67100 L’Aquila, Italy; 5Department of Medicine and Surgery, University of Insubria, 21100 Varese, Italy; 6Department of Medicine and Cardiopulmonary Rehabilitation, Maugeri Care and Research Institutes, IRCCS Tradate, 21100 Varese, Italy; 7Biomedical Department of Internal Medicine and Specialistics, University of Palermo, 90100 Palermo, Italy; 8Clinica Medica, Department of Medicine and Surgery, University of Milano-Bicocca, 20900 Monza, Italy; 9Studium Patavinum, Department of Medicine, University of Padua, 35100 Padua, Italy; 10Department of Translational and Precision Medicine, Sapienza University of Rome, 00185 Rome, Italy; 11Department of Advanced Biomedical Sciences, “Federico II” University of Naples, 80133 Naples, Italy; 12Department Hypertension and Cardiovascular Disease Research Center, Medical and Surgical Sciences Department, Alma Mater Studiorum University of Bologna, 40126 Bologna, Italy; 13Heart-Chest-Vascular Department, IRCCS AOU of Bologna, 40126 Bologna, Italy; 14Department of Public Health, “Federico II” University of Naples, 80133 Naples, Italy; 15Nephrology, Dialysis and Transplantation Unit, Department of Emergency and Organ Transplantation, “Aldo Moro” University of Bari, 70122 Bari, Italy; 16Department of Clinical Medicine and Surgery, “Federico II” University of Naples, 80133 Naples, Italy; 17Cardiology IV, “A.De Gasperi’s” Department, Niguarda Ca’ Granda Hospital, 20162 Milan, Italy; 18School of Medicine and Surgery, Milano-Bicocca University, 20126 Milan, Italy; 19Italian Society of General Medicine (SIMG), 67051 Avezzano, Italy; 20CNR-IFC, Clinical Epidemiology of Renal Diseases and Hypertension, Reggio Cal Unit, 89124 Reggio Calabria, Italy; 21Department of Internal Medicine, Santa Maria Della Misericordia General Hospital, AULSS 5 Polesana, 45100 Rovigo, Italy; 22Department of Clinical and Experimental Sciences, University of Brescia, 25121 Brescia, Italy; 23Department of Precision and Regenerative Medicine and Jonic Area (DiMePRe-J), Neurosciences and Sense Organs, University of Bari Medical School, 70122 Bari, Italy; 24S. Luca Hospital, Istituto Auxologico Italiano & University of Milan-Bicocca, 20126 Milan, Italy; 25Department of Internal Medicine, University of Genoa; IRCSS Ospedale Policlinico San Martino, 16132 Genova, Italy; 26Department of Medicine—DIMED, University of Padova, Medicina Interna 1°, Ca’ Foncello University Hospital, 31100 Treviso, Italy; 27Department of Medical and Surgical Science, University of Perugia, 06100 Perugia, Italy; 28Department of Geriatric and Intensive Care Medicine, Careggi Hospital, University of Florence, 50121 Florence, Italy; 29Department of Medicine, University of Padua, 35100 Padua, Italy; 30Department of Clinical and Molecular Medicine, University of Rome Sapienza, 00185 Rome, Italy; 31Hospital S. Maria della Misericordia, 06100 Perugia, Italy

**Keywords:** serum uric acid, triglycerides, cardiovascular, risk prediction, mortality, cardiometabolic, hypertriglyceridemia

## Abstract

High serum uric acid (SUA) and triglyceride (TG) levels might promote high-cardiovascular risk phenotypes across the cardiometabolic spectrum. However, SUA predictive power in the presence of normal and high TG levels has never been investigated. We included 8124 patients from the URic acid Right for heArt Health (URRAH) study cohort who were followed for over 20 years and had no established cardiovascular disease or uncontrolled metabolic disease. All-cause mortality (ACM) and cardiovascular mortality (CVM) were explored by the Kaplan–Meier estimator and Cox multivariable regression, adopting recently defined SUA cut-offs for ACM (≥4.7 mg/dL) and CVM (≥5.6 mg/dL). Exploratory analysis across cardiometabolic subgroups and a sensitivity analysis using SUA/serum creatinine were performed as validation. SUA predicted ACM (HR 1.25 [1.12–1.40], *p* < 0.001) and CVM (1.31 [1.11–1.74], *p* < 0.001) in the whole study population, and according to TG strata: ACM in normotriglyceridemia (HR 1.26 [1.12–1.43], *p* < 0.001) and hypertriglyceridemia (1.31 [1.02–1.68], *p* = 0.033), and CVM in normotriglyceridemia (HR 1.46 [1.23–1.73], *p* < 0.001) and hypertriglyceridemia (HR 1.31 [0.99–1.64], *p* = 0.060). Exploratory and sensitivity analyses confirmed our findings, suggesting a substantial role of SUA in normotriglyceridemia and hypertriglyceridemia. In conclusion, we report that SUA can predict ACM and CVM in cardiometabolic patients without established cardiovascular disease, independent of TG levels.

## 1. Introduction

Cardiovascular (CV) disease remains the leading cause of morbidity and mortality worldwide [1]. Indeed, the prevalence of cardiometabolic diseases (obesity, arterial hypertension, and diabetes mellitus) is steadily increasing [2], which in turn increases the incidence of end-stage CV syndromes such as heart failure [3]. Indeed, despite significant improvement in pharmacological treatment, cardiometabolic diseases are characterised by a high residual CV risk [4,5]. Their high morbidity and mortality rates burden national healthcare systems [1]. Therefore, prevention [6] or early intensive treatment of the disease [7,8] represent the best intervention strategies. Identifying CV risk factors that might be informative of early cardiovascular derangement is an unmet critical need [9,10].

Serum uric acid (SUA) and triglycerides (TG) [11,12], which are closely related [13], are involved in early cardiometabolic damage [14]. Due to conflicting evidence in clinical trials, they have long been neglected in CV risk stratification, especially when compared with other lipid profile components (e.g., LDL and non-HDL cholesterol) [15,16]. In recent years, however, several observations [17], Mendelian randomizations [18], and meta-analyses [19,20] have shown that SUA is a predictor of CV disease in the general population. More importantly, its impact on CV risk starts at a lower level than the traditional threshold for defining hyperuricemia and gout [21,22]. In recent years, the URic acid Right for heArt Health (URRAH) study has identified specific cut-offs for all-cause mortality (ACM) and cardiovascular mortality (CVM) to support its implementation in clinical practice [11,23,24,25,26,27,28]. Similarly, recent real-world evidence supports the association of TG with CV risk starting from levels defining mild to moderate hypertriglyceridemia (150 mg/dL) [29,30], and Mendelian randomization studies also support a causal role for TG in CV disease [31]. Initial controversial results may also partly depend on the heterogeneity of the study populations, particularly with regard to their metabolic phenotype: as elevated SUA and TG levels are common features of metabolic diseases [14], their potential association with CV events may simply reflect the presence of impaired metabolic control.

Furthermore, some reports suggest a possible interplay between SUA and TG [32]. They may share molecular damage pathways, contributing to the activation of NRLP3-inflammasome [33], which characterises the metainflammatory state of cardiometabolic disease [34]. As low-grade inflammation is involved in early and metabolic-related CV damage [35], SUA and TG may be particularly informative in risk stratification in cardiometabolic patients without established CV disease. The relatively neglected role of TG in cardiovascular risk stratification [15,16], its tight relationship with SUA, and their potential molecular interaction make investigating their potential interplay substantially relevant. However, the predictive role of SUA and TG in cardiometabolic patients has yet to be explored.

We aimed to evaluate the impact of SUA in terms of ACM and CVM in cardiometabolic patients without established CV disease in conditions of normal and high TG levels. To provide tangible results and rapid implementation into the clinical routine, we used a pragmatic approach by adopting acknowledged SUA and TG cut-offs for ACM and CVM.

## 2. Materials and Methods

### 2.1. Protocol Design

The URRAH study is a multicentre observational cohort study involving patients consecutively referred to different centres for the diagnosis, treatment, and management of arterial hypertension across the Italian territory. The total follow-up period is 133 (63–153) months, up to 31 July 2017. The study protocol has been previously described in detail [11,36]. Briefly, a nationwide Italian database was constructed by collecting individual data on patients with SUA measurements and anthropometric and biochemical characterisation, and clinical history information. The eGFR was calculated according to the CKD-EPI formula [37]. Study data were collected routinely or ad hoc in previously approved studies. Participants underwent no extra tests nor interventions, and there was no impact on participants’ care or outcome. The URRAH study was conducted in accordance with the Declaration of Helsinki for Human Research. The processing of the patients’ personal data collected in this study complied with the European Directive on the Privacy of Data. Approval was sought from the Ethical Committee of the Coordinating Center at the Division of Internal Medicine of the University of Bologna (No. 77/2018/Oss/AOUBo). All participants signed informed consent for participation in the study and publication of the results.

### 2.2. Adoption of Specific SUA and TG Cut-Offs

We designed the sub-analysis to investigate the impact of SUA on ACM and CVM in cardiometabolic patients without established CV disease across normal and high TG levels. We followed a pragmatic approach, using established cut-offs to define diseases and pathological conditions. Higher levels of SUA were defined as ≥4.7 mg/dL when assessing ACM and ≥5.6 mg/dL when assessing CVM. Hypertriglyceridemia (hTG) was defined as TG ≥ 150 mg/dL, according to the current ESC guidelines on cardiovascular prevention [6].

### 2.3. Cardiometabolic Disease Definitions

Obesity was defined as BMI ≥ 30 kg/m^2^. Hypertension (HT) was defined by at least two blood pressure (BP) recordings ≥ 140 mmHg for systolic, ≥90 mmHg for diastolic [6], or treatment with antihypertensive medications. Diabetes mellitus (DM) was defined according to ADA criteria [38]. Patients were considered healthy if they were referred to our outpatient clinics for preliminary screening to rule out hypertension, did not fall into any of the previous definitions, and were not receiving any cardiometabolic-related medication.

### 2.4. Study Population Selection from the National Regional-Based Dataset

A flowchart of the sub-analysis is reported in Figure 1. The URRAH project database included a total of 27,078 patients. People with missing data on any of the parameters of interest for the statistical analysis (duration of follow-up time, survival status, CVM status, diabetes mellitus, hypertension, SUA, fasting TG, age, sex, BMI, alcohol consumption, smoking, systolic BP (SBP), diastolic BP (DBP), total cholesterol, HDL, fasting blood glucose, creatinine, hematocrit, and diuretics) were excluded from the analysis (n = 15,278). To focus on the early-to-moderate cardiometabolic disease, we further excluded all the patients with either an established diagnosis of CV disease or a biochemical profile suggestive of uncontrolled disease (n = 3676): previous CV events, history of heart failure, SBP > 240 mmHg, DBP > 140 mmHg, fasting blood glucose > 350 mg/dL, or serum creatinine > 4 mg/dL. A total of 8124 patients were included in the final analysis.

### 2.5. Outcomes

ACM and CVM were assessed using data from hospital records or death certificates. Mortality from major CV disease (International Classification of Diseases, Tenth Revision) included deaths from diseases of the heart, essential hypertension, hypertensive renal disease, and cerebrovascular diseases, as previously reported [11].

### 2.6. Statistical Analysis

Continuous variables were tested for normality by Shapiro–Wilk tests. Data were presented as mean ± SD for continuous parametric variables, median (1st–3rd quartile) for continuous non-parametric variables and percentages for binary variables. Non-normal variables were natural-log-transformed when used in parametric tests. The national regional-based dataset was then stratified according to the condition of hTG versus normal triglyceridemia (nTG). Pearson correlation was used to assess association. The independent sample Student’s *t*-test was adopted to explore differences between groups, and the χ^2^ test and Fisher’s exact test for categorical data, as appropriate. The national regional-based dataset was further stratified according to SUA cut-offs for ACM and CVM for survival analysis. Kaplan–Meier survival curves, log-rank tests, and Cox proportional-hazards models were used to analyse the association between baseline SUA and the time-to-event (ACM and CVM). Data were censored at the time of the last visit or, for patients lost during follow-up, at the last date they were known to be alive. Associations are expressed as hazard ratio (HR) [95% confidence interval]. Multivariable Cox regression models were designed to include all available clinical variables with biological plausibility to ensure consistency with previous URRAH studies [11]. Specifically, the variables considered potential confounders were: age, sex, BMI, active alcohol consumption, current smoking habit, SBP, DBP, total cholesterol, HDL, glycaemia, creatinine, hematocrit, and use of diuretics. A value of *p* < 0.05 was considered statistically significant. For the means of an exploratory analysis across the cardiometabolic spectrum, patients were further stratified according to cardiometabolic condition: the presence of no disease (healthy), single disease (Ob, HT, DM), or sum of disease (Ob + HT, Ob + DM, HT + DM, Ob + HT + DM). The analysis was repeated within each subgroup of patients with different cardiometabolic disorders. A subsequent analysis with simultaneous stratification for SUA and TG in four groups (low SUA and nTG: lSUA_nTG; high SUA and nTG: hSUA_nTG; low SUA and hTG: lSUA_hTG; high SUA and hTG: hSUA_hTG) was performed to further explore the interaction between the two parameters across the cardiometabolic spectrum. Finally, a sensitivity analysis was performed adopting the SUA/serum creatinine ratio (SUA/sCr) instead of SUA, with a cut-off point of 5.35 for both ACM and CVM [25]. All analyses were performed using jamovi (The Jamovi project 2022, version 2.3.18).

## 3. Results

### 3.1. Baseline Characterisation of the Study Population Selected from the National Regional-Based Dataset

The general characterisation of the study population as a whole and according to nTG vs. hTG is shown in Table 1. Patients with hTG (n = 1865, 23% of the total) were older, with higher BMI, BP, fasting plasma glucose, and hematocrit. They had a worse lipid profile, a more prevalent smoking and alcohol habit, and a higher burden of comorbidities. SUA and TG levels were related in the whole cohort (*p* < 0.001, r = 0.332) and across TG strata (*p* < 0.001, r = 0.262 for nTG; *p* < 0.001, r = 0.149 for hTG) (Figure 2). During a median follow-up of 130 (50–156) months, n = 1740 (21.4%) ACM events and n = 840 (10.3%) CVM events were observed.

### 3.2. The Predictive Power of SUA Cut-Offs in Normotriglyceridemia and Hypertriglyceridemia

Cox regression analysis (Table 2) confirmed that both SUA and TG above cut-off values were associated with an increased risk of ACM: HR 1.25 [1.12–1.40] (*p* < 0.001) and HR 1.24 [1.09–1.39] (*p* = 0.001), respectively. However, only SUA cut-offs were able to predict CVM: 1.31 [1.11–1.74] (*p* < 0.001). Interaction terms (SUA strata*TG strata) were not significant in either analysis. Also, no difference was observed between the biological sexes.

Stratification for TG showed a higher number of ACM events in the hTG group: 465 (24.9%) vs. 1275 (20.4%), *p* < 0.001. Similarly, the number of CVM events was higher in the hTG group: 229 (12.3%) vs. 611 (9.8%), *p* = 0.002. The Kaplan–Meier curve analysis showed a distinct separation of the curves since the beginning, with SUA cut-offs predicting ACM and CVM both in nTG and hTG (*p* < 0.001 each) (Figure 3). Cox regression analysis confirmed that the SUA cut-off was reliable for predicting ACM in nTG (HR 1.26 [1.12–1.43], *p* < 0.001) and hTG (1.31 [1.02–1.68], *p* = 0.033). The SUA cut-off for CVM (≥5.6 mg/dl), in turn, discriminated well between high and low risk in participants with nTG (HR 1.46 [1.23–1.73], *p* < 0.001), while it almost reached significance in the hTG group after adjustment (HR 1.31 [0.99–1.64], *p* = 0.060) (Table 3).

### 3.3. Predictive Value of SUA across the Cardiometabolic Spectrum

The subgroup exploration across different facets of the cardiometabolic health-to-disease transition (healthy subjects, patients with obesity, patients with hypertension, patients with diabetes, and patients with a cumulative burden of pathologies) showed that the SUA cut-offs had a consistent positive trend in terms of higher ACM and CVM risk prediction (Appendix A, Figure 4) in univariable and multivariable Cox regression analyses. In particular, for ACM, SUA ≥ 4.7 mg/dL was significantly related to higher risk in nTG healthy individuals (1.55 [1.09–2.21], *p* < 0.015), in nTG patients with obesity (5.79 [1.20–27.98], *p* = 0.029), and in nTG hypertensive patients (1.28 [1.09–1.52]. *p* = 0.002). For CVM, SUA ≥ 5.6 mg/dL predicted the event in nTG patients with hypertension (1.37 [1.08–1.74], *p* = 0.010), in nTG patients with diabetes (6.92 [1.50–31.91], *p* = 0.013), and in nTG patients with hypertension and diabetes (1.60 [1.02–2.53], *p* = 0.043). The same exploration performed for combined TG and SUA strata confirmed the results, showing that high SUA is a robust independent predictor of ACM and CVM in the early cardiometabolic spectrum (Appendix A).

### 3.4. Sensitivity Analysis Adopting SUA/Serum Creatinine Ratio

To further validate our results, the same analysis was repeated by adopting the SUA/sCr ratio instead of SUA [25]. The Kaplan–Meier curve analysis and the Cox regression analysis confirmed the predictive power of the SUA/sCr ratio in the study population (Table 4). In both TG strata (Figure 5, Table 5), in the cardiometabolic subgroups (Appendix A) and in the cardiometabolic subgroups after combined TG and SUA strata (Appendix A), the SUA/sCr cut-off was more efficient in predicting ACM and CVM in patients without hypertriglyceridemia.

## 4. Discussion

In the present study, we report for the first time that: (a) SUA cut-offs recently identified by our group [11], as well as hypertriglyceridemia [29,30], predict ACM and CVM in cardiometabolic patients without established CV disease; (b) SUA and TG have an independent effect on mortality risk prediction; (c) the risk prediction of SUA cut-offs is confirmed even after stratification for TG levels; (d) the exploratory and sensitivity analyses show that the specific cut-offs are consistent in normotriglyceridemic conditions at the early stages of the cardiometabolic spectrum. Elevated SUA and TG levels have been increasingly investigated as potential novel predictors and risk factors for all-cause and cardiovascular mortality. Low-cost biochemical parameters become highly relevant in the clinical setting when they can discriminate early cardiometabolic phenotypes at a higher risk of mortality. Given (i) the easy availability of SUA testing, also in the primary care setting, (ii) the confirmation of recently acknowledged cut-off values (SUA ≥ 4.7 mg/dL for ACM and SUA ≥ 5.6 mg/dL for CVM), far below those defining hyperuricemia [11], and (iii) the ability to predict risk from the early stages of cardiometabolic damage [39], our work supports the immediate inclusion of SUA in routine clinical practice to promote proactive risk stratification and cardiovascular prevention strategies.

In recent years, several papers from our group [11,23,24,25,26,27,28,40,41,42,43,44] and others [17,18,19,20] have reported that SUA might be a relevant tool in the risk-prediction arsenal for assessing individual mortality risk at values lower than the gout-defining ones. However, fewer studies have attempted to determine the potential of SUA as a risk factor for mortality and cardiovascular disease in a population at an early stage of cardiometabolic impairment. Chang et al. showed that n = 973 non-hypertensive non-diabetic subjects in the highest tertile for SUA (≥6.1 mg/dL) had significantly higher Framingham Risk Score (FRS) [45]. In n = 12,637 adults with obesity, a nonlinear relationship was reported between SUA (with an adopted cut-off of 6.5 mg/dL) and ACM, but not with CVM [46]. In n = 667 patients with hypertension, high SUA (≥9.0 mg/dL) was found to be associated with ACM and CVM [47]. However, these works provide higher and different cut-offs, hindering their adoption in routine clinical care. After having identified two lower and specific cut-offs for ACM and CVM (SUA ≥ 4.7 mg/dL for ACM and SUA ≥ 5.6 mg/dL for CVM) [11], our group has shown their consistent predictive power in patients with diabetes [23] and metabolic syndrome [24]. The present study confirms the findings in cardiometabolic patients without established CV disease and considers the potential simultaneous assessment of hypertriglyceridemia, another neglected potential CV risk factor [15,16]. It also confirms that hypertriglyceridemia, defined using the lower cut-off accepted in the literature [6], predicts mortality in patients with a mild-to-moderate cardiometabolic burden.

The relevance of investigating the predictive potential of SUA in conditions of normo- and hypertriglyceridemia lies in the potential interplay between these two cardiovascular risk factors. Several epidemiological observations have reported an association between SUA and TG levels [13,48,49], also focusing on cardiometabolic conditions [14,50]; this relationship was confirmed in our study. However, no studies have explored the predictive role of SUA in terms of mortality risk across TG strata. We herein report for the first time that SUA retains its ability to discriminate between higher and lower ACM and CVM risk independently from the TG levels. Curiously, this is different from what we recently found for HDL, suggesting distinct interplays between SUA and different components of the lipid profile [51].

Further investigation of the interplay between these two CV risk factors led to peculiar findings. First, our exploratory analyses across the cardiometabolic spectrum highlighted how SUA cut-offs are stronger predictors of ACM and CVM in healthy individuals or in patients with a mild cardiometabolic burden. Although this issue has been little explored in the literature, Yang et al. interestingly reported that the association they found between SUA and left ventricular diastolic dysfunction was stronger in subjects with no other cardiometabolic risk factors [52]. The work from Chiang et al., showing that hyperuricemia is, per se, a very early cardiometabolic disturbance, might indirectly support our findings [53]. Furthermore, our exploratory and sensitive analyses showed that the predictive role of the recently accepted SUA cut-offs seems to be higher in patients with normal triglyceride levels. As some evidence suggests a possible interaction between uric acid and triglycerides [32,33], these results might seem controversial. However, several explanations might be adduced. The smaller sample size of the hypertriglyceridemic part of the study population selected from the regional-based dataset, and even more so of some specific subgroups, may have hindered the assessment of the effect. Also, we did not investigate the use of different cut-offs for both SUA and TG. However, according to our pragmatic approach, this allows our results to be immediately implemented in a clinical setting. Finally, the common pathways between uric acid and triglycerides might be characterised by a threshold effect. SUA and TG might thus saturate the low-grade inflammatory response by competitively activating molecular pathways as NLRP3 [33], so the combination of both factors turns out to be non-additive in terms of mortality. This might also explain the discrepancies observed when adopting SUA/sCr rather than SUA. As SUA/sCr includes renal function and might be a better CV risk discriminant than SUA alone [25], the saturation of common damage pathways might clarify why SUA/sCr confirms its predictive role only in the condition of normotriglyceridemia. Further studies should be undertaken to clarify this specific issue.

Our work has several limitations. First, the URRAH study cohort includes patients consecutively referred to different centres for the diagnosis, treatment, and management of arterial hypertension across the Italian territory. Thus, although it also has non-hypertensive patients (due to rule-out after preliminary screening), patients with arterial hypertension represent the greater portion of the study cohort, which could represent a selection bias. Second: partly as a consequence, the small sample size of some subgroups included in the exploratory analysis across the cardiometabolic spectrum may have limited the appreciation of the predictive power. However, particularly for normotriglyceridemic subjects, the consistency of the effect across the early spectrum and the confirmation of the results when adopting SUA/sCr instead of SUA support the validity of the findings. Third, although we hypothesise that a possible explanation of the non-significant effect of SUA in subjects with hypertriglyceridemia could be due to a saturation of the low-grade inflammatory response, because high-sensitivity C-reactive protein levels are only available for a very limited number of subjects, it was not possible to include them in the analysis. Fourth, some other relevant variables were missing in the URRAH dataset. In particular, we could not perform adjustments for glycated haemoglobin, a crucial parameter when assessing glucose control. Fifth, we did not investigate other cut-off values for SUA in our study. However, we deliberately took a pragmatic approach and conducted our work to take advantage of the potential role of defined accepted cut-offs from other recent studies from our group. This indeed provides consistency with previous work and conveys a clear clinical message to the clinician, i.e., the validity of SUA cut-offs of 4.7 mg/dL and 5.6 mg/dL as predictors of ACM and CVM, respectively, making our findings of rapid and easy clinical implementation.

## 5. Conclusions

In the present study, we have, for the first time, investigated the predictive power of recently defined SUA cut-offs for mortality (all-cause and cardiovascular) in cardiometabolic subjects with and without hypertriglyceridemia and without established CV disease. We report here that SUA levels above 4.7 mg/dL and 5.6 mg/dL are consistently able to predict mortality (all-cause and cardiovascular) and that their risk stratification ability appears to be particularly relevant in early cardiometabolic disease in patients without hypertriglyceridemia. Tailored and very early cardiovascular prevention is essential to reduce the cardiometabolic burden that characterises the 21st-century healthcare systems worldwide [1,54]. As SUA is a widely available and inexpensive biochemical parameter, even in the primary care setting, our observation supports a proactive approach to early cardiovascular risk stratification. Furthermore, it opens up intriguing speculative hypotheses about the mechanistic interplay between uric acid and triglycerides in cardiometabolism, which should be addressed in further studies.

## Figures and Tables

**Figure 1 metabolites-13-00244-f001:**
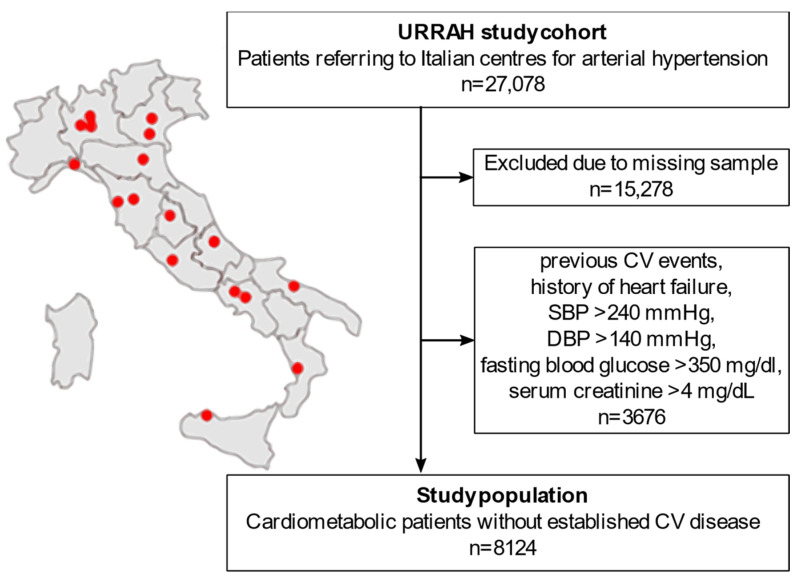
Flowchart of the study population selection from the national regional-based dataset. CV: cardiovascular. SBP: systolic blood pressure. DBP: diastolic blood pressure. URRAH: URic acid Right for heArt Health.

**Figure 2 metabolites-13-00244-f002:**
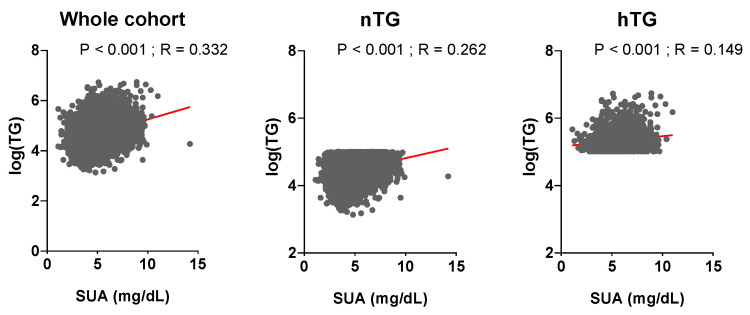
Relationship between uric acid and triglyceride levels. Serum uric acid and triglyceride levels are correlated (*p* ≤ 0.001, St.β 0.26, r = 0.332). Triglyceride levels (mg/dL) are log-transformed. Spearman coefficient was used to assess correlation. *p* < 0.05 was deemed statistically significant. *TG: triglycerides. SUA: serum uric acid.*

**Figure 3 metabolites-13-00244-f003:**
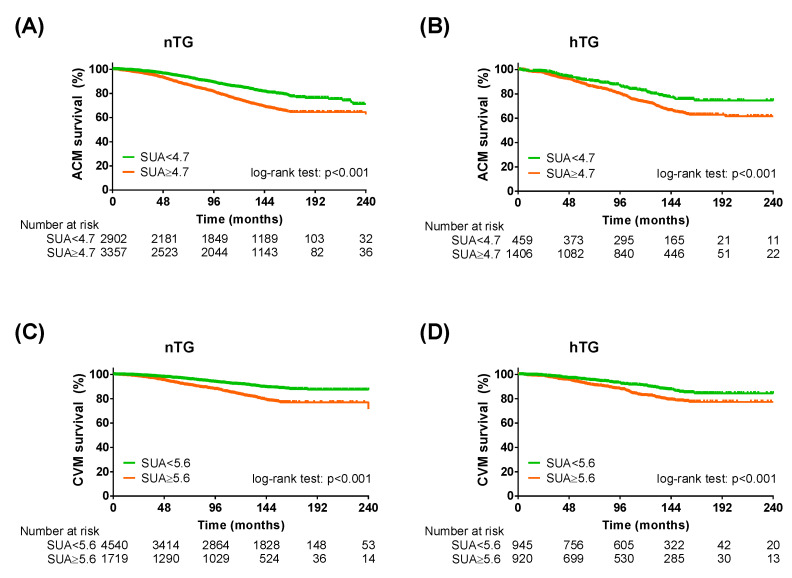
Kaplan–Meier survival analysis across triglycerides strata. Kaplan–Meier curves for SUA cut-offs in normotriglyceridemia for all-cause mortality (**A**) and cardiovascular mortality (**C**) and KM curves for SUA cut-offs in hypertriglyceridemia for all-cause mortality (**B**) and cardiovascular mortality (**D**). SUA < 4.7 mg/dL (for all-cause mortality) or <5.6 mg/dL (for cardiovascular mortality): green lines; SUA ≥ 4.7 mg/dL (for all-cause mortality) or ≤5.6 mg/dL (for cardiovascular mortality): orange lines. Log-rank test was used to compare the curves. *p* < 0.05 was deemed statistically significant. *nTG: normotriglyceridemia. hTG: hypertriglyceridemia. SUA: serum uric acid.*

**Figure 4 metabolites-13-00244-f004:**
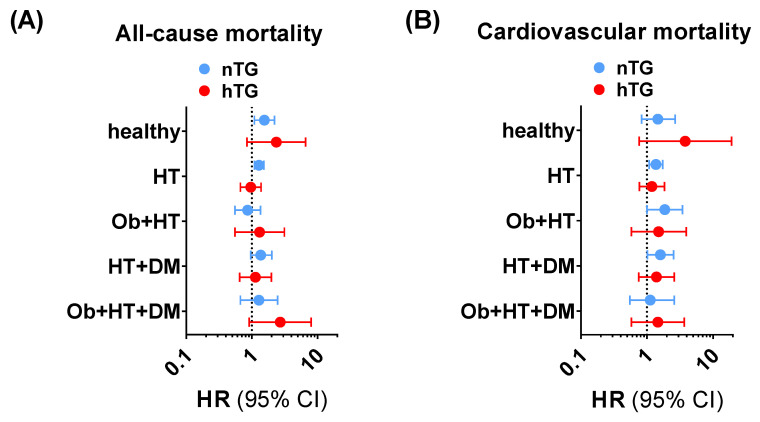
Trends for all-cause mortality (**A**) and cardiovascular mortality (**B**) across the cardiometabolic spectrum in patients with normotriglyceridemia (light blue dots and lines) vs. hypertriglyceridemia (red dots and lines) adopting SUA ≥ 4.7 mg/dL and SUA ≥ 5.6 mg/dL as cut-offs for all-cause mortality and cardiovascular mortality, respectively. The analysis was not run on people with obesity and type diabetes and no other comorbidities due to the small sample size of the subgroup (n = 18). Data in undersized groups (n < 50) are reported for completeness in Appendix A. *CI: confidence interval. DM: patients with diabetes and no other comorbidities. HR: hazard ratio. HT: patients with hypertension and no other comorbidities. HT + DM: patients with hypertension and diabetes and no other comorbidities. Ob: patients with obesity and no other comorbidities. Ob + HT: patients with obesity and hypertension and no other comorbidities. Ob + HT + DM: patients with obesity, hypertension, and diabetes, and no other comorbidities.*

**Figure 5 metabolites-13-00244-f005:**
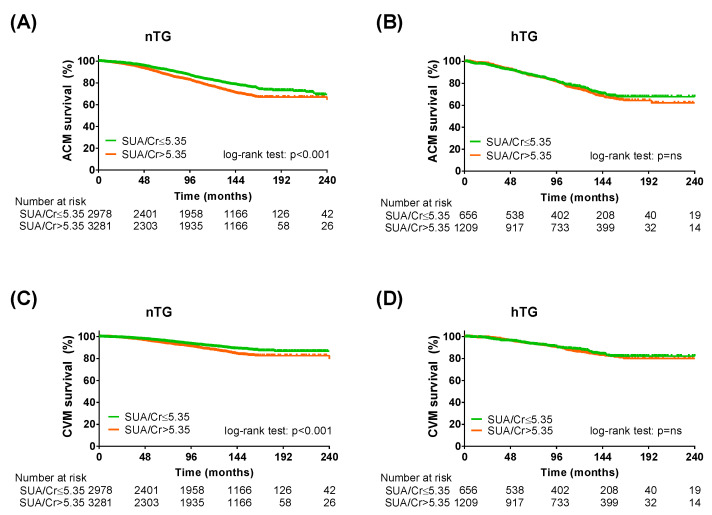
Kaplan–Meier survival analysis across triglycerides strata adopting SUA/serum creatinine cut-off. Kaplan–Meier curves for SUA cut-offs (SUA/creatinine ≤ 5.35: green lines; SUA/creatinine > 5.35: orange lines) in normotriglyceridemia for all-cause mortality (**A**) and cardiovascular mortality (**C**) and KM curves for SUA cut-offs in hypertriglyceridemia for all-cause mortality (**B**) and cardiovascular mortality (**D**). Log-rank test was used to compare the curves. *p* < 0.05 was deemed statistically significant. *Cr: serum creatinine; nTG: normotriglyceridemia. hTG: hypertriglyceridemia. SUA: serum uric acid.*

**Table 1 metabolites-13-00244-t001:** Characteristics of the study population selected from the national regional-based dataset.

	Whole Dataset (n = 8124)	nTG (n = 6259)	hTG (n = 1865)	*p*-Value
**Age** (years)	58.8 ± 16.1	58.3 ± 16.5	60.4 ± 14.9	<0.001
**Male sex** (%)	43.8	44.2	42.3	0.139
**BMI** (kg/m^2^)	26.2 ± 4.2	25.7 ± 4.1	27.8 ± 4.0	<0.001
**Smokers** (%)	20.3	19.0	24.5	<0.001
**Alcohol habit** (%)	66.6	65.8	69.4	0.004
**Systolic blood pressure** (mmHg)	148 ± 25	146 ± 26	153 ± 24	<0.001
**Diastolic blood pressure** (mmHg)	88 ± 12	87 ± 12	91 ± 11	<0.001
**Uric acid** (mg/dL)	5.0 ± 1.4	4.9 ± 1.3	5.7 ± 1.4	<0.001
**Total cholesterol** (mg/dL)	213 ± 40	208 ± 39	231 ± 37	<0.001
**HDL cholesterol** (mg/dL)	55.1 ± 15.4	57.7 ± 15.3	55.1 ± 15.4	<0.001
**Triglycerides** (mg/dL)	104 (76–145)	90 (69–144)	190 (167–237)	<0.001
**Fasting plasma glucose** (mg/dL)	94 (86–105)	93 (84–102)	99 (90–110)	<0.001
**eGFR** (mL/min/1.73 m^2^)	85 ± 20	87 ± 20	80 ± 20	<0.001
**Hematocrit** (%)	42.5 ± 3.8	42.3 ± 3.8	43.3 ± 3.8	<0.001
**Obesity** (%)	16.5	13.8	25.5	<0.001
**Arterial hypertension** (%)	73	69.7	84.0	<0.001
**Type 2 diabetes** (%)	10.8	8.9	16.9	<0.001
**Diuretics** (%)	15.6	14.3	20.2	<0.001

Baseline characteristics of the study population as a whole and after TG strata. Data are reported as mean ± standard deviation or median (1st–3rd quartile), as appropriate. Non-normal variables were natural-log-transformed when used in parametric tests. Data were compared by independent sample Student’s *t*-test (continuous data) and the χ^2^ test (categorical data). *p* < 0.05 was considered statistically significant. *eGFR: estimated glomerular filtration rate. hTG: hypertriglyceridemia. nTG: normotriglyceridemia.*

**Table 2 metabolites-13-00244-t002:** Cox regression analysis in the study population selected from the national regional-based dataset for SUA and TG cut-offs.

All-Cause Mortality				
	Univariable	*p*-value	Multivariable	*p*-value
SUA ≥ 4.7	1.79 [1.61–1.98]	<0.001	1.25 [1.12–1.40]	<0.001
TG ≥ 150	1.25 [1.12–1.39]	<0.001	1.24 [1.09–1.39]	0.001
**Cardiovascular Mortality**				
	Univariable	*p*-value	Multivariable	*p*-value
SUA ≥ 5.6	2.03 [1.77–2.32]	<0.001	1.31 [1.11–1.74]	<0.001
TG ≥ 150	1.28 [1.10–1.49]	<0.001	1.13 [0.95–1.39]	0.163

Hazard ratios [95% confidence interval] for SUA and TG cut-offs in univariable and multivariable analysis for all-cause and cardiovascular mortality. In the multivariable model, both the TG and SUA cut-offs were included. Age, sex, BMI, the active consumption of alcohol, a current smoking habit, SBP, DBP, total cholesterol, HDL, glycaemia, creatinine, hematocrit, and diuretics were used as confounders. Interaction terms (SUA strata*TG strata) were found non-significant. Data were analysed by Cox regression analysis. *p* < 0.05 was considered statistically significant. *TG: triglycerides. SUA: serum uric acid.*

**Table 3 metabolites-13-00244-t003:** Cox regression analysis for SUA cut-offs across TG strata.

All-Cause Mortality (SUA ≥ 4.7)				
	Univariable	*p*-value	Multivariable	*p*-value
nTG	1.31 [1.18–1.47]	<0.001	1.26 [1.12–1.43]	<0.001
hTG	1.60 [1.26–2.02]	<0.001	1.31 [1.02–1.68]	0.033
**Cardiovascular Mortality (SUA ≥ 5.6)**				
	Univariable	*p*-value	Multivariable	*p*-value
nTG	2.13 [1.81–2.50]	<0.001	1.46 [1.23–1.73]	<0.001
hTG	1.66 [1.28–2.17]	<0.001	1.31 [0.99–1.64]	0.060

Hazard ratios [95% confidence interval] for SUA cut-offs across TG strata in univariable and multivariable analysis for all-cause and cardiovascular mortality. Age, sex, BMI, the active consumption of alcohol, a current smoking habit, SBP, DBP, total cholesterol, HDL, glycaemia, creatinine, hematocrit, and diuretics were used as confounders. Data were analysed by Cox regression analysis. *p* < 0.05 was considered statistically significant. *hTG: hypertriglyceridemia. nTG: normotriglyceridemia. SUA: serum uric acid.*

**Table 4 metabolites-13-00244-t004:** Cox regression analysis in the study population selected from the national regional-based dataset for SUA/serum creatinine and TG cut-offs.

All-Cause Mortality				
	Univariable	*p*-value	Multivariable	*p*-value
SUA/serum creatinine > 5.35	1.33 [1.21–1.47]	<0.001	1.18 [1.07–1.31]	0.001
TG ≥ 150	1.25 [1.12–1.39]	<0.001	1.29 [1.14–1.45]	<0.001
**Cardiovascular Mortality**				
	Univariable	*p*-value	Multivariable	*p*-value
SUA/serum creatinine > 5.35	1.37 [1.19–1.58]	<0.001	1.19 [1.03–1.37]	0.016
TG ≥ 150	1.28 [1.10–1.49]	0.001	1.21 [1.02–1.44]	0.028

Hazard ratios [95% confidence interval] for SUA/serum creatinine and TG cut-offs in univariable and multivariable analysis for all-cause and cardiovascular mortality. In the multivariable model, both the TG and SUA/serum creatinine cut-offs were included. Age, sex, BMI, the active consumption of alcohol, a current smoking habit, SBP, DBP, total cholesterol, HDL, glycaemia, hematocrit, and diuretics were used as confounders. Interaction terms (SUA/serum creatinine strata*TG strata) were found non-significant. Data were analysed by Cox regression analysis. *p* < 0.05 was considered statistically significant. *TG: triglycerides. SUA: serum uric acid.*

**Table 5 metabolites-13-00244-t005:** Cox regression analysis in the study population selected from the national regional-based dataset for SUA/serum creatinine cut-offs across TG strata.

All-Cause Mortality				
	Univariable	*p*-value	Multivariable	*p*-value
nTG	1.31 [1.18–1.47]	<0.001	1.47 [1.30–1.66]	<0.001
hTG	1.10 [0.90–1.33]	0.348	1.09 [0.89–1.32]	0.404
**Cardiovascular Mortality**				
	Univariable	*p*-value	Multivariable	*p*-value
nTG	1.44 [1.23–1.69]	<0.001	1.23 [1.04–1.45]	0.015
hTG	1.08 [0.82–1.42]	0.578	1.09 [0.82–1.44]	0.548

Hazard ratios [95% confidence interval] for SUA/serum creatinine cut-off across TG strata in univariable and multivariable analysis for all-cause and cardiovascular mortality. Age, sex, BMI, the active consumption of alcohol, a current smoking habit, SBP, DBP, total cholesterol, HDL, glycaemia, hematocrit and diuretics were used as confounders. Data were analysed by Cox regression analysis. *p* < 0.05 was considered statistically significant. *hTG: hypertriglyceridemia. nTG: normotriglyceridemia. SUA: serum uric acid.*

## Data Availability

Data are available upon reasonable request to the investigators. The data are not publicly available due to specific restriction (as data under Ethical Committee protection).

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
