# Peer review of "Serum Uric Acid Predicts All-Cause and Cardiovascular Mortality Independently of Hypertriglyceridemia in Cardiometabolic Patients without Established CV Disease: A Sub-Analysis of the URic acid Right for heArt Health (URRAH) Study"

_metabolites, 2023, doi:10.3390/metabo13020244_

Round 1
Reviewer 1 Report
Manuscript “Serum uric acid predicts all-cause and cardiovascular mortality independently of hypertriglyceridemia in cardiometabolic patients without established CV disease: a sub-analysis of the URic acid Right for heArt Health (URRAH) study” is a very interesting work. It shows, that serum uric acid can predict all-cause mortality and cardiovascular mortality in cardiometabolic patients without established cardiovascular disease, independent of triglycerides levels.
However, I believe there are some problems that need to be corrected.
1. Table formatting is incorrect.
2. There are no footnotes under Table 1.
3. It must be stated that the values presented in the table are the mean with standard deviation.
4. References prepared not in accordance with the guidelines.
Author Response
RE: metabolites-2205686, entitled “Serum uric acid predicts all-cause and cardiovascular mortality independently of hypertriglyceridemia in cardiometabolic patients without established CV disease: a sub-analysis of the URic acid Right for heArt Health (URRAH) study”.
Reviewer comments to the Authors.
Reviewer #1: Manuscript “Serum uric acid predicts all-cause and cardiovascular mortality independently of hypertriglyceridemia in cardiometabolic patients without established CV disease: a sub-analysis of the URic acid Right for heArt Health (URRAH) study” is a very interesting work. It shows, that serum uric acid can predict all-cause mortality and cardiovascular mortality in cardiometabolic patients without established cardiovascular disease, independent of triglycerides levels.
However, I believe there are some problems that need to be corrected.
Response: We would like to thank the Reviewer for the positive and constructive comments on our work.
- Table formatting is incorrect.
Response: We apologise for this inaccuracy. We have now formatted the Tables according to the journal requirements.
- There are no footnotes under Table 1. It must be stated that the values presented in the table are the mean with standard deviation.
Response: Thanks for noticing this flaw: we have now included Table 1 footnotes in the manuscript (lines 252-257).
- References prepared not in accordance with the guidelines.
Response: The referee is correct, and we apologise for this inaccuracy. We have now formatted the references according to the MDPI style.

Reviewer 2 Report
Authors investigated whether SUA predicts ACM and CVM, independently of TG status, in patients with cardiometabolic diseases without CVD. Their findings are interesting, and the paper is well written. This reviewer has several comment to be addressed.
Comment 1. Figure 1: The meaning of ‘non-endstage’ is unclear for me.
Comment 2. Confirm, fasting TG?
Comment 3. T2D seem to be identified presence or absence according to ADA criteria only. How about just having T2D and being considered in patients with ‘uncontrolled metabolic disease’? Did authors consider the glycemic status, such as HbA1c?
Comment 4. Authors focus on TG in the usual lipid parameters. Please provide a more detailed rationale why TG rather than LDL or non-HDL was chosen as the lipid parameter for the analysis.
Comment 5. Figure 2: Need units after logarithmic conversion??
Comment 6. Table 2: ‘No difference was observed between biological sex (Table 2)’ Where in Table 2?
Comment 7. Tables 3 and 5: Need P values for interaction between nTG vs. hTG.
Comment 8. Figure 4: Please add ‘Ob (only)’ and ‘DM (only)’.
Comment 9. Discussion on the difference in the impact of TG levels between SUA (Figure 3) and SUA/Cr (Figure 5).
Author Response
RE: metabolites-2205686, entitled “Serum uric acid predicts all-cause and cardiovascular mortality independently of hypertriglyceridemia in cardiometabolic patients without established CV disease: a sub-analysis of the URic acid Right for heArt Health (URRAH) study”.
Reviewer comments to the Authors.
Reviewer #2: Authors investigated whether SUA predicts ACM and CVM, independently of TG status, in patients with cardiometabolic diseases without CVD. Their findings are interesting, and the paper is well written. This reviewer has several comment to be addressed.
Response: We would like to thank the Reviewer for the positive and constructive comments on our work.
- Figure 1: The meaning of ‘non-endstage’ is unclear for me.
Response: The reviewer is correct that “non-endstage” might seem confusing. Thus, we have now corrected Figure 1, replacing “patients with non-endstage cardiometabolic disease” with “Cardiometabolic patients without established cardiovascular disease”, consistent with the title and the text.
- Confirm, fasting TG?
Response: We confirm that triglyceride levels were measured at fasting. We have better specified this in the methods section when referring to the variables adopted in the analysis (line 182).
- T2D seem to be identified presence or absence according to ADA criteria only. How about just having T2D and being considered in patients with ‘uncontrolled metabolic disease’? Did authors consider the glycemic status, such as HbA1c?
Response: T2D was identified based on a clinical diagnosis according to ADA criteria. Thus, T2D does not translate into having an uncontrolled metabolic disease. Indeed, as the Reviewer suggests, having an abnormal glycaemic value could indicate an underlying uncontrolled disease. Accordingly, we had included the glycemic status when assessing disease control to exclude extremely deranged subjects (line 188). However, glycated haemoglobin values were not available in our study database. Thus, we included them neither in the selection criteria for uncontrolled disease nor in the analysis. We have now acknowledged this in the limitations (lines 451-453).
- Authors focus on TG in the usual lipid parameters. Please provide a more detailed rationale why TG rather than LDL or non-HDL was chosen as the lipid parameter for the analysis.
Response: We agree that LDL and non-HDL cholesterol are lipid parameters which have been widely explored in terms of association with cardiovascular risk prediction. Indeed, they have also been extensively investigated (notably, also by our group [1]). However, we focused on TG for several reasons:
- Hypertriglyceridemia and hyperuricemia are parts of the early cardiometabolic continuum [2], and are tightly related [3] (lines 108-110, 121-125).
- Both triglycerides and serum uric acid have been partially neglected as cardiovascular risk predictors in the past; however, over the last years, several observations have started to take both of them into higher consideration when assessing cardiovascular risk [4-10] (lines 110-121).
- Triglycerides and serum uric acid might share common pathways of damage in the context of metabolic disease, as suggested by a recent work focusing on the NRLP3-inflammasome [11] (lines 126-130).
We have extensively discussed this in the introduction. Also, according to the Reviewer’s suggestion, we have further commented on the neglected role of triglycerides compared to usual lipid parameters (lines 110-111) and summarised all the above points to convey a more solid rationale (lines 131-135).
- Figure 2: Need units after logarithmic conversion??
Response: We agree with the Reviewer that Figure 2 might seem confusing. We have now amended it and implemented the figure caption to clarify (line 261).
- Table 2: ‘No difference was observed between biological sex (Table 2)’ Where in Table 2?
Response: We did not observe sex-specific differences related to TG strata; we did not report them in Table 2 to avoid data overcrowding. However, we agree with the Reviewer that the sentence might sound confounding. We have now rephrased the paragraph (lines 246-250).
- Tables 3 and 5: Need P values for interaction between nTG vs. hTG.
Response: No interaction terms between nTG and hTG have been investigated in Tables 3 and 5, as we meant to explore the predictive power of SUA cut-offs within distinct sub-groups. However, in the whole study population, the independent predictive power of nTG vs hTG has been investigated (Tables 2 and 4), as well as its interaction with SUA cut-offs and was found non-significant, as it was reported in the text (lines 249-250, 343-345).
- Figure 4: Please add ‘Ob (only)’ and ‘DM (only)’.
Response: for data clarity, we would avoid inserting the above-mentioned subgroups, given the difficult graphical representation of their confidence intervals due to the very small sample size of these two groups. However, we have included them in Supplementary Tables 1 and 2 (as also reported in the Figure 4 caption, lines 321-322) for completeness.
- Discussion on the difference in the impact of TG levels between SUA (Figure 3) and SUA/Cr (Figure 5).
Response: Thanks for pointing out this relevant point, which allows us to discuss the different pathophysiologic and predictive meanings of the SUA/sCr with respect to SUA alone. Indeed, we have now commented on this extensively on the paper (lines 431-435).
References
- Palatini, P.; Virdis, A.; Masi, S.; Mengozzi, A.; Casiglia, E.; Tikhonoff, V.; Cicero, A.F.G.; Ungar, A.; Parati, G.; Rivasi, G., et al. Hyperuricemia increases the risk of cardiovascular mortality associated with very high hdl-cholesterol level. Nutr Metab Cardiovasc Dis 2022.
- Lurbe, E.; Torro, M.I.; Alvarez-Pitti, J.; Redon, J.; Borghi, C.; Redon, P. Uric acid is linked to cardiometabolic risk factors in overweight and obese youths. J Hypertens 2018, 36, 1840-1846.
- Wang, X.; Zhong, S.; Guo, X. The associations between fasting glucose, lipids and uric acid levels strengthen with the decile of uric acid increase and differ by sex. Nutr Metab Cardiovasc Dis 2022, 32, 2786-2793.
- Selvaraj, S.; Claggett, B.L.; Pfeffer, M.A.; Desai, A.S.; Mc Causland, F.R.; McGrath, M.M.; Anand, I.S.; van Veldhuisen, D.J.; Kober, L.; Janssens, S., et al. Serum uric acid, influence of sacubitril-valsartan, and cardiovascular outcomes in heart failure with preserved ejection fraction: Paragon-hf. Eur J Heart Fail 2020, 22, 2093-2101.
- Kleber, M.E.; Delgado, G.; Grammer, T.B.; Silbernagel, G.; Huang, J.; Krämer, B.K.; Ritz, E.; März, W. Uric acid and cardiovascular events: A mendelian randomization study. J Am Soc Nephrol 2015, 26, 2831-2838.
- Zhao, G.; Huang, L.; Song, M.; Song, Y. Baseline serum uric acid level as a predictor of cardiovascular disease related mortality and all-cause mortality: A meta-analysis of prospective studies. Atherosclerosis 2013, 231, 61-68.
- Gill, D.; Cameron, A.C.; Burgess, S.; Li, X.; Doherty, D.J.; Karhunen, V.; Abdul-Rahim, A.H.; Taylor-Rowan, M.; Zuber, V.; Tsao, P.S., et al. Urate, blood pressure, and cardiovascular disease: Evidence from mendelian randomization and meta-analysis of clinical trials. Hypertension 2021, 77, 383-392.
- Toth, P.P.; Fazio, S.; Wong, N.D.; Hull, M.; Nichols, G.A. Risk of cardiovascular events in patients with hypertriglyceridaemia: A review of real-world evidence. Diabetes Obes Metab 2020, 22, 279-289.
- Tan, H.L.E.; Hure, A.; Peel, R.; Hancock, S.; Attia, J. Prevalence and clinical risk prediction of hypertriglyceridaemia in a community cohort. Intern Med J 2021.
- Holmes, M.V.; Asselbergs, F.W.; Palmer, T.M.; Drenos, F.; Lanktree, M.B.; Nelson, C.P.; Dale, C.E.; Padmanabhan, S.; Finan, C.; Swerdlow, D.I., et al. Mendelian randomization of blood lipids for coronary heart disease. Eur Heart J 2015, 36, 539-550.
- Schunk, S.J.; Kleber, M.E.; März, W.; Pang, S.; Zewinger, S.; Triem, S.; Ege, P.; Reichert, M.C.; Krawczyk, M.; Weber, S.N., et al. Genetically determined nlrp3 inflammasome activation associates with systemic inflammation and cardiovascular mortality. Eur Heart J 2021, 42, 1742-1756.

Reviewer 3 Report
In this work, the investigators evaluate the impact of serum uric acid in relation to cardiovascular mortality and all-cause mortality in a subgroup of patients from a larger cohort; they also evaluate the impact of triglyceride levels.
The work seems to me to be adequately written and presented. Among its strengths is the large number of study subjects, although when performing the sub-analyses some study groups are considerably reduced. In addition, the fact that the data come from a cohort is another of its strengths.
Although there are some similar works on this topic, the analysis of the predictive power, the cut-off point for uric acid levels, and the relative ease of practical application of the results makes the manuscript of interest to a broad group of health care professionals.
However, it is worth noting the high percentage of self-citations, more than 10 percent.
Author Response
RE: metabolites-2205686, entitled “Serum uric acid predicts all-cause and cardiovascular mortality independently of hypertriglyceridemia in cardiometabolic patients without established CV disease: a sub-analysis of the URic acid Right for heArt Health (URRAH) study”.
Reviewer comments to the Authors.
Reviewer #3: In this work, the investigators evaluate the impact of serum uric acid in relation to cardiovascular mortality and all-cause mortality in a subgroup of patients from a larger cohort; they also evaluate the impact of triglyceride levels.
The work seems to me to be adequately written and presented. Among its strengths is the large number of study subjects, although when performing the sub-analyses some study groups are considerably reduced. In addition, the fact that the data come from a cohort is another of its strengths.
Although there are some similar works on this topic, the analysis of the predictive power, the cut-off point for uric acid levels, and the relative ease of practical application of the results makes the manuscript of interest to a broad group of health care professionals.
However, it is worth noting the high percentage of self-citations, more than 10 percent.
Response: We are very grateful to the Reviewer for the positive comments on our work. We are completely in line with the potential application to healthcare professionals, starting from primary care physicians. Concerning self-citations, we acknowledged the point, and we agree with the Reviewer; however, our group has extensively focused on SUA’s predictive role in terms of cardiovascular disease over the last year, struggling to identify lower and more suitable cut-offs easily transferable to the clinical setting, which is the reason behind the several papers from the same group cited by the present work.

Round 2
Reviewer 2 Report
Thank you for your hard revising during the short period.
I have no further comments.